# Magnetic eye tracking in mice

Hannah L Payne, Jennifer L Raymond*

Department of Neurobiology, Stanford University, Stanford, United States

**Abstract** Eye movements provide insights about a wide range of brain functions, from sensorimotor integration to cognition; hence, the measurement of eye movements is an important tool in neuroscience research. We describe a method, based on magnetic sensing, for measuring eye movements in head-fixed and freely moving mice. A small magnet was surgically implanted on the eye, and changes in the magnet angle as the eye rotated were detected by a magnetic field sensor. Systematic testing demonstrated high resolution measurements of eye position of <0.1°. Magnetic eye tracking offers several advantages over the well-established eye coil and video-oculography methods. Most notably, it provides the first method for reliable, high-resolution measurement of eye movements in freely moving mice, revealing increased eye movements and altered binocular coordination compared to head-fixed mice. Overall, magnetic eye tracking provides a lightweight, inexpensive, easily implemented, and high-resolution method suitable for a wide range of applications.
DOI: https://doi.org/10.7554/eLife.29222.001

## Introduction

Many behaviors, from navigation and predation to avoidance of danger and pursuit of mates, are guided by vision (*Chapillon, 1999*; *Coen and Murthy, 2016*; *Mather and Baker, 1980*; *Olberg et al., 2000*; *Yilmaz and Meister, 2013*). Vision is actively controlled by eye movements, which localize and stabilize images on the retina. Therefore, eye movements are often measured in studies of visually guided behaviors, both for documentation and control of the visual inputs imping-ing on the retina, and as a behavioral output. The analysis of eye movements can provide a window on a variety of sensory, motor, and cognitive functions, such as attention, learning and memory, and decision-making (*Duhamel et al., 1992*; *Kustov and Robinson, 1996*; *Raposo et al., 2012*; *Raymond and Lisberger, 1996*). Moreover, abnormal eye movements are associated with numerous cognitive disorders, including autism, schizophrenia, Parkinson's disease, and Huntington's disease (*Avanzini et al., 1979*; *DeJong and Jones, 1971*; *Holzman et al., 1976*; *Rosenhall et al., 1988*). Thus, the measurement of eye movements can advance our understanding of the sophisticated neu-ral processes underlying a wide range of functions in healthy and diseased brains.

Given their broad utility, measurements of eye movements have been routinely implemented in neuroscientific studies of primates. In contrast, such measurements are uncommon in mice, a species that is widely used in molecular and genetic studies of behavior (*Beery and Zucker, 2011*). Although less dependent on vision than primates, mice encode visual input with high selectivity (*Niell and Stryker, 2008*), and use vision for a variety of functions including navigation (*Chapillon, 1999*; *Chen et al., 2013a*; *Hopkins, 1953*; *Saleem et al., 2013*), avoidance of predators (*Yilmaz and Meis-ter, 2013*), and recognition of territorial boundaries (*Mackintosh, 1973*). Eye movements can alter the visual experience during these behaviors, and thus provide key information about how these behaviors are controlled. However, measurement of eye movements in mice is technically challenging.

The main obstacle to measuring eye movements in mice is their diminutive size. In particular, the small eye restricts what can be implanted on the eye's surface: average eye diameter is only 3.4 mm in mice (*Remtulla and Hallett, 1985*), compared to 6.4 mm in rat (*Chaudhuri et al., 1983*), 9.2 mm

*For correspondence:
jenr@stanford.edu

in chicken (*Schwarz et al., 2013*), 18 mm in rabbit (*Hughes, 1972*), and 21 mm in rhesus macaque (*Schultz, 1940*). The small size of the head and body also impede measurement of eye movements, especially during free motion, by restricting what can be affixed to the skull and carried by the mouse. Whereas neuroscientific studies of awake, behaving primates are typically conducted in head-fixed animals, behavioral paradigms in mice often leverage locomotion (*Fyhn et al., 2008*; *Machado et al., 2015*; *Moser et al., 2008*; *Yoder and Taube, 2009*), nose pokes (*Raposo et al., 2012*), foraging (*Devito and Eichenbaum, 2011*; *Jennings et al., 2015*), social interaction (*Silverman et al., 2010*), or other complex behaviors that require freedom of motion. The power of such behavioral paradigms has motivated significant efforts to analyze head and body kinematics (*Fyhn et al., 2008*; *Goulding et al., 2008*; *Hardcastle et al., 2017*; *Machado et al., 2015*; *Wiltschko et al., 2015*; *Yoder and Taube, 2009*) and to image the activity of neural populations (*Barretto et al., 2009*; *Chen et al., 2013b*; *Flusberg et al., 2008*; *Jennings et al., 2015*) in freely moving mice. However, measurements of eye movements have been lacking in these studies, even for visually guided behaviors such as navigation. Thus, a method for analyzing eye movements in unrestrained mice would be a powerful tool for advancing our understanding of the murine behavioral repertoire and its neural underpinnings. Previously, the only eye tracking method compatible with free motion in small animals was the electro-oculogram, which is susceptible to inaccuracies (*Arden and Kelsey, 1962*). Recently, video-oculography was used to measure visually driven eye movements in mice without head restraint, however the mice were constrained to a small platform, and only 20% of the measurements were useable (*Kretschmer et al., 2017*). Here, we address the need for an accurate and reliable technique to measure eye movements in mice during the broad range of freely moving behaviors used in neuroscience research.

Our approach was designed to overcome the major limitations of previous eye tracking methods in mice, the most popular of which are the eye coil method and video-oculography. The eye coil, or scleral search coil, technique is considered the gold standard for measuring eye movements in most species (*Boyden and Raymond, 2003*; *Collewijn et al., 1975*; *Judge et al., 1980*; *Koekkoek et al., 1997*; *Remmel, 1984*; *Robinson, 1963*). A small coil of wire is surgically implanted beneath the conjunctiva on the surface of the eye, and eye position is determined from the current generated in the wire by an external magnetic field. A major advantage of the eye coil method is that it has high spatial (~0.1°) and temporal (<1 ms) resolution (*Collewijn et al., 1975*; *Stahl et al., 2000*). Disadvantages arise from the difficulty of implementation, particularly in species with small eyes. The surgery requires lengthy training to master, and the fragile wires leading subcutaneously from the eye coil to a skull-mounted connector can break, requiring additional surgeries for repair or additional animals. Moreover, during measurements of eye movements, it is optimal for the animal's head to remain stationary and centered within large magnetic field coils, so this approach has been applied almost exclusively in head-fixed animals (but see *Collewijn, 1977*; *Ogorodnikov et al., 2006*; *Sánchez-López and Escudero, 2015*).

A second, common method for measuring eye movements is video-oculography, which uses a high-speed camera to record the position of the pupil (*Mitchiner et al., 1976*; *Stahl et al., 2000*; *Wallace et al., 2013*). The primary advantage of this approach is that it is non-invasive. However, there are also several disadvantages. Its spatial resolution is lower than that of the eye coil (*Houben et al., 2006*; *Kimmel et al., 2012*; *McCamy et al., 2015*; *Stahl et al., 2000*). Inaccuracies can arise from changes in pupil diameter due to fluctuations in luminance or behavioral state (*Kaufman, 2002*; *Kimmel et al., 2012*; *McCamy et al., 2015*; *Reimer et al., 2014*). A camera or mirror must be placed in the animal's field of view, which limits the experimenter's control of visual stimuli. Finally, the head must remain stationary relative to the camera, which precludes use in freely moving mice, since even the miniaturized video-oculography system developed for the rat (*Wallace et al., 2013*) significantly exceeds the size of the mouse head.

A promising alternative to the eye coil and video-oculography techniques is an approach based on magnetic sensing. If a magnet is implanted on the surface of the eye, its magnetic field will rotate with the eye, allowing eye movements to be detected with a sensor that measures either the angle or the strength of the magnetic field. Three previous reports, two in goldfish (*Rodríguez et al., 2001*; *Salas et al., 1999*) and one in chickens (*Schwarz et al., 2013*) have demonstrated the potential of magnetic sensing for measuring eye movements. We extended these initial efforts in several ways. First, we miniaturized the implant for the smaller mouse eye, while achieving spatial resolution comparable to the eye coil benchmark. Second, we rigorously tested robustness to the relative

placement of the magnet and sensor. Third, we developed an improved method for calibrating the magnetic eye tracking system. These advances yielded the first high resolution recordings of eye movements in freely moving mice.

## Results

We developed a system for measuring eye movements in mice using magnetic sensing technology. A powerful neodymium magnet was surgically implanted beneath the conjunctiva so that it rotated with the eye, and a magnetic field sensor was used to detect the resulting changes in the angle of the magnetic field.

### Magnetic sensor performance outside the animal

The performance of the magnetic eye tracking system was first tested outside the mouse, where the position of the magnet and sensor could be precisely controlled, allowing systematic comparison of different magnet-sensor alignments. Initially, a 0.75 × 2 mm cylinder magnet was positioned directly below and adjacent to the sensor, with the N-S axis of the magnet parallel to the plane of the sensor, and the magnet's axis of rotation aligned to the center of the sensor (*Figure 1A*, *left*, with δ = 0 mm). When the magnet was rotated ±90° about an axis perpendicular to the plane of the sensor, the magnetic sensor produced a voltage that depended on the angle of the magnet (*Figure 1A*, *center*, *black traces*).

The magnetic sensor we used has two channels, which relay output from two sets of magnetoresistive elements at a 45° orientation relative to each other, so that the position tuning of the two channels is offset by 45° (*Figure 1A*, *center*). Hence, during normal eye movements in mice, which are typically within ±10–20° (*van Alphen et al., 2010*), at least one of the two channels should operate close to its range of greatest sensitivity and linearity (near 0° for Channel 1, and 45° for Channel 2 in *Figure 1A*, *center*).

Robustness to deviations in magnet-sensor alignment is necessary for magnetic eye tracking in vivo, since surgical implantation of the magnet and sensor will inevitably result in some variation in placement. Most notably, the magnet cannot be directly adjacent to the sensor when implanted in the mouse eye. Therefore, we tested the effect of increasing the distance to the magnet on the sensitivity of the sensor to the angular position of the magnet. Sensitivity was calculated as the slope of the relationship between magnetic sensor voltage and angular position of the magnet, averaged in a ±22.5° range centered on the zero-crossings of each channel. Sensitivity decreased from 224.7 ± 0.3 mV/° at 0 mm to 28.1 ± 1.0 mV/° at 5 mm (average sensitivity of the two channels; sensitivity for each channel is shown separately in *Figure 1A*, *right*). Notably, at 3 mm, a distance readily achievable in mice (see Materials and methods), the sensitivity was 127.0 ± 1.2 mV/°. This corresponded to a spatial resolution, as measured by the standard deviation of the signal at rest, of 0.016° ± 0.005°, which is nearly an order of magnitude below the noise floor measured by the best eye tracking methods in mice (*Stahl et al., 2000*). Results obtained using a 1.5 × 0.5 mm disc magnet were similar to those for the cylinder magnet, with a sensitivity of 90.9 ± 0.7 mV/° at a distance of 3 mm (*Figure 1—figure supplement 1*). Because of its marginally greater sensitivity, we focused on the cylinder magnet for all subsequent measurements.

In addition to characterizing how sensitivity varied with the vertical distance between the magnet and sensor, the effect of horizontal distance (displacement within a plane parallel to the plane of the sensor) was tested. In the animal, the magnet will not rotate exactly about its center, but instead will be horizontally offset by the radius of the eye. We tested the impact of this offset by fixing the vertical distance between magnet and sensor at 3 mm, and then horizontally displacing the magnet from the midpoint of the sensor and from its axis of rotation (*Figure 1B*). Sensitivity decreased with distance; nevertheless, the signals were still more than half maximum at 1.5 mm of horizontal offset (*Figure 1B*), which corresponds to the radius of the mouse eye. When the sensor, rather than the magnet, was displaced from the axis of rotation, sensitivity decreased in a similar manner (*Figure 1C*). Therefore, the sensor's ability to detect magnet angle is robust to horizontal as well as vertical variations in magnet-sensor alignment of magnitudes that would be expected in vivo.

Finally, we tested the effect of tilting the N-S axis of the magnet relative to the plane of the sensor, which also may occur during implantation. Tilting the magnet by 45° did not affect sensitivity (127.8 ± 4 mV/° no tilt, 128.0 ± 2.0 mV/° with 45° tilt, *Figure 1D*).

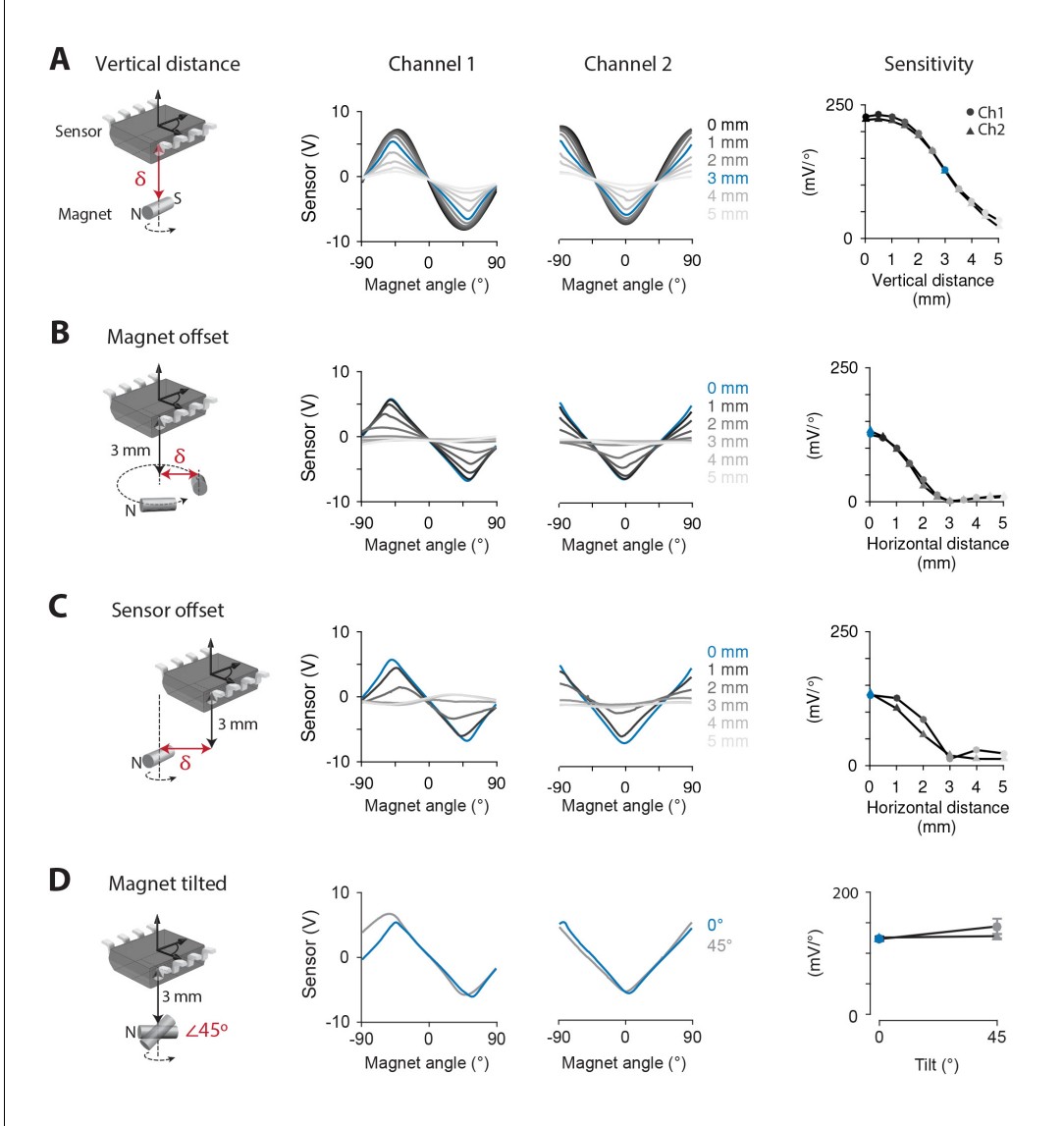

**Figure 1.** Systematic testing of magnet-sensor placement. (A) Output of the magnetic sensor when a 0.75 × 2 mm cylinder magnet was rotated by a servo-controlled motor. *Left,* Schematic showing the relative position and orientation of the magnet and sensor, the axis of magnet rotation (*dashed line and arrow*) and the dimension along which magnet position was varied ($\delta$, *red arrow*). *Middle,* Output from each channel of the magnetic sensor as vertical distance ($\delta$) between the magnet and the surface of the sensor was varied from 0 mm (*black*) to 5 mm (*light grey*) (n = 5 repeated measurements). *Right,* Sensitivity of each channel of the magnetic sensor at each distance from the magnet. In this and all panels, SEM for repeated measurements was smaller than the symbol size. (B) Same as in (A), but the magnet was offset horizontally ($\delta$) from both its axis of rotation and the center of the sensor. Vertical distance was fixed at 3 mm. (C) Same as in (B), but the sensor was offset horizontally from the center of the magnet, which rotated about its center. (D) Same as in (B), but comparing horizontal orientation of the magnet as in (A) with a 45° tilt relative to the plane of the sensor (p=0.168, n = 5 repeated measurements, two sample *t*-test). Vertical distance between the center of the magnet and the sensor was fixed at 3 mm.

DOI: https://doi.org/10.7554/eLife.29222.002

The following figure supplement is available for figure 1:

**Figure supplement 1.** Systematic testing of magnet-sensor placement for a disc magnet.

DOI: https://doi.org/10.7554/eLife.29222.003

## Magnetic tracking of eye movements in mice

After characterizing the performance of the magnetic eye tracking system outside the mouse, we implemented magnetic eye tracking in head-fixed mice. A cylinder magnet was surgically implanted beneath the conjunctiva on the temporal margin of one eye, with the N-S axis aligned so that it

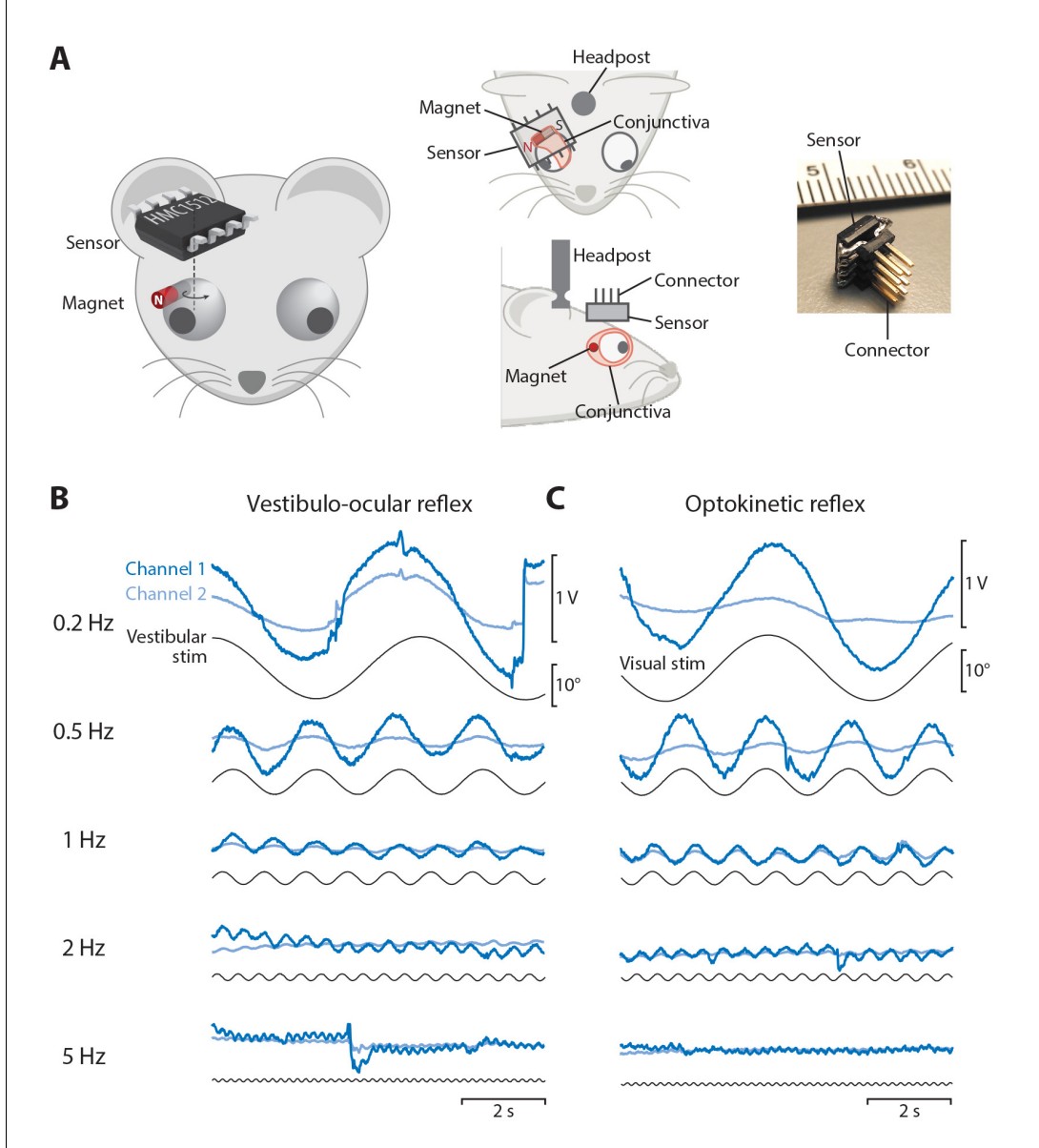

**Figure 2.** Magnetic eye tracking in mice. (**A**) Schematic of the magnetic eye tracking system, illustrating the orientation of the sensor relative to the magnet and eye. The magnet was implanted beneath the conjunctiva on the temporal side of the eye. The sensor was soldered to an 8-pin connector and positioned over the magnet in a plane parallel to that of horizontal (nasal-temporal) eye movements. An implanted head post was used to restrain the head so that the nasal-temporal axis of the eye was earth-horizontal. (**B**) Raw output from both channels of the magnetic sensor (*blue traces*) during the vestibulo-ocular reflex (VOR) in one example mouse. A vestibular stimulus (*black trace*, head position) was delivered by rotating the animal about an earth-vertical axis in the dark (0.2 Hz to 5 Hz, peak velocity ±10°/s). (**C**) Raw output from the magnetic sensor during the optokinetic response (OKR). A visual stimulus (*black trace*) was delivered by rotating a vertically striped drum, which surrounded the mouse, about an earth-vertical axis (0.2 Hz to 5 Hz, peak velocity ±10°/s).
DOI: https://doi.org/10.7554/eLife.29222.004

created a rotating magnetic field during horizontal (nasal-temporal) eye movements (*Figure 2A*; see Materials and methods). A magnetic sensor was affixed to the skull in a plane roughly parallel to the nasal-temporal axis of the eye. The head was restrained, via a surgically implanted head post, with the nasal-temporal axis of the eye in an earth-horizontal plane.

The ability of the magnetic eye tracking system to detect eye movements was tested by delivering vestibular and visual stimuli known to evoke eye movements in head-fixed mice. The vestibulo-ocular reflex (VOR) drives compensatory eye movements in response to passive head rotation in the dark, in mice (*Iwashita et al., 2001*; *Katoh et al., 1998*; *Kimpo et al., 2005*; *van Alphen et al., 2001*) as in other species. Accordingly, when vestibular stimuli were delivered to mice implanted with the magnetic eye tracking system, the magnetic sensor detected robust eye movements driven by the VOR. Sinusoidal vestibular stimuli at a range of stimulus frequencies from 0.2 Hz to 5 Hz elicited sinusoidal eye movements at the corresponding frequencies (*Figure 2B*).

The magnetic eye tracking approach was also effective at detecting visually-driven eye movements in head-fixed mice. The optokinetic reflex (OKR) drives image-stabilizing eye movements in response to motion of a visual stimulus, in mice (*Iwashita et al., 2001*; *Katoh et al., 1998*) as in other species. Accordingly, when a striped drum was rotated around the mouse, the magnetic sensor detected robust eye movements driven by the OKR, at all stimulus frequencies tested (0.2 Hz–5 Hz; *Figure 2C*). Thus, the magnetic eye tracking system can detect both vestibularly- and visually-driven eye movements in head-fixed mice.

## Calibration of magnetic eye tracking using dual-angle video-oculography

To convert the raw output of the magnetic sensor (in volts) to eye position (in degrees), we developed an inexpensive yet accurate calibration system based on video-oculography. A major motivation for developing magnetic eye tracking in mice was to overcome the limitations of video-oculography, such as its limited spatial resolution and the incompatibility with free motion of the animal, which restrict its use for certain applications. However, these limitations do not preclude the use of video-oculography for calibration of the magnetic sensor.

We modified a standard video-oculography technique to simplify implementation and improve accuracy. In species other than primates, eye position is typically computed from video measurements using an estimate of $R_p$, the distance of the pupil from the center of corneal curvature (*Figure 3A*; *Stahl et al., 2000*). However, difficulty in estimating $R_p$ and variations in $R_p$ across individuals and during the course of an experiment are sources of inaccuracy (*Kaufman, 2002*; *Kimmel et al., 2012*; *McCamy et al., 2015*; *Reimer et al., 2014*; *Wisard et al., 2010*; *Zoccolan et al., 2010*). We eliminated the need to estimate $R_p$ by using simultaneous image capture of the eye from two cameras. By positioning the cameras at a known, fixed angle relative to each other (40°, in our setup) and extracting the locations of the pupil and a reference corneal reflection (CR) in each camera's image, the angle of the eye can be calculated using simple trigonometric identities (*Figure 3A,B*; see Materials and methods). To validate the dual-angle video-oculography system, two mice were anesthetized and the cameras were rotated at known angles around one eye, over a ± 25° range. The eye position measured with video-oculography agreed closely with true position over the full range (*Figure 3C*). Thus, dual-angle video-oculography accurately reports eye position, and hence can be used to calibrate the signals recorded by the magnetic sensor.

To calibrate the signals recorded by the magnetic sensor in each mouse, eye movements were simultaneously recorded in awake, head-restrained mice using the video-oculography and magnetic eye tracking systems. During a brief (1–3 min) calibration session, 0.5 Hz or 1 Hz sinusoidal vestibular stimuli were used to elicit eye movements (*Figure 4A*). A calibration factor for the magnetic sensor was extracted by using linear regression to predict video-derived horizontal eye position from the raw voltages of the magnetic sensor. The resulting calibration factor, in units of °/mV, could then be used to calculate eye position from the signals recorded by the magnetic sensor.

The signal in one channel of the magnetic sensor was often considerably larger and more strongly correlated with the video-derived measurements of eye position than the signal in the other channel (see examples in *Figure 2B,C* and *Figure 4A*), as expected from tests outside the animal (*Figure 1*). Nonetheless, the second channel might carry some additional information about eye position. Therefore, we assessed whether a better estimate of eye position could be obtained from a linear model incorporating both channels of the magnetic sensor, rather than the single best channel (the channel with the highest correlation coefficient to video-derived eye position, see Materials and methods). On average, incorporating both channels only improved the fit to the video-derived measurements by a small amount (3.5% additional variance explained, *Figure 4B,C*). We also estimated eye position using a model that included a nonlinear, quadratic term for each channel, and found little

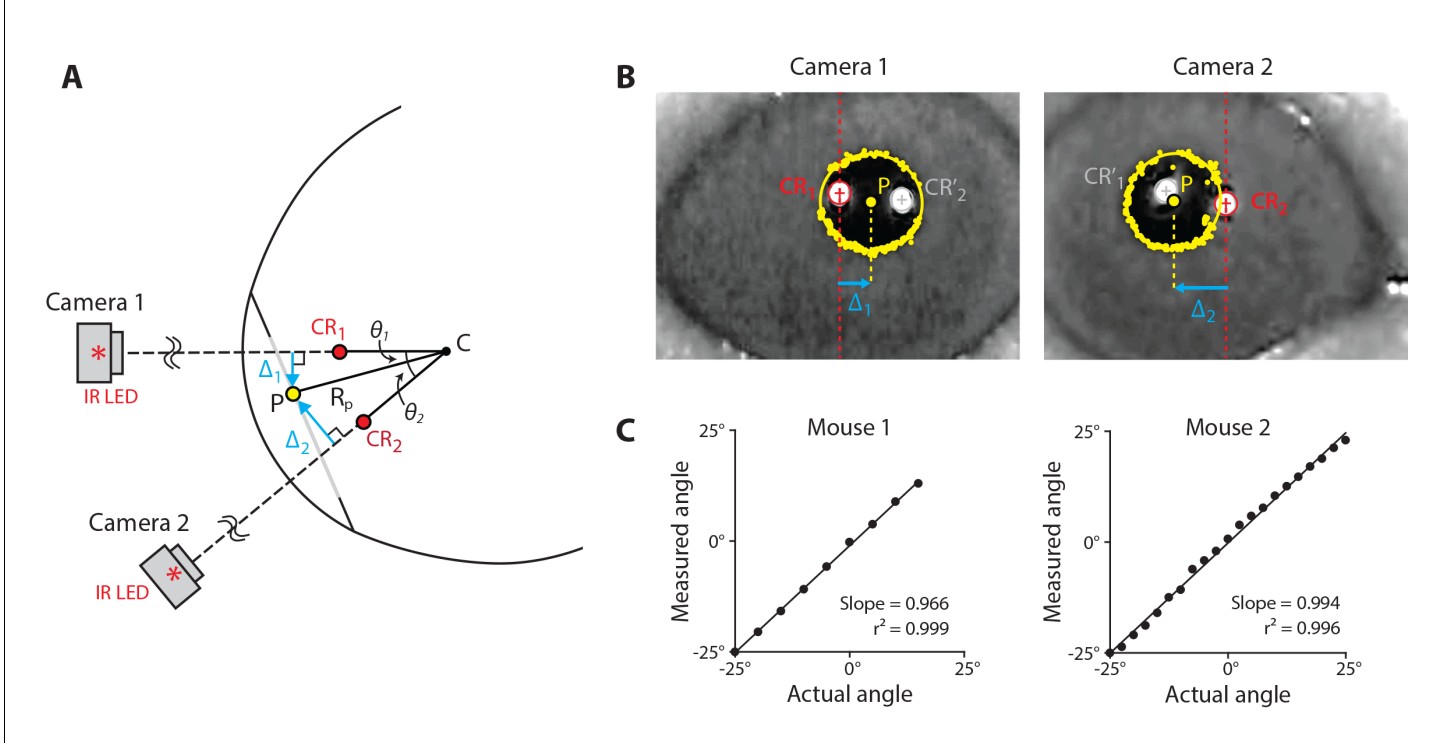

**Figure 3.** Dual-angle video-oculography. (**A**) Schematic of the eye as viewed from above, illustrating the dual-angle video-oculography technique. Two cameras were affixed to a platform, with their axes at an angle of 40° relative to each other, and with the cameras equidistant (5 cm) from the point at which their axes intersected. An infrared LED mounted directly above each camera created a corneal reflection (CR). The position of the camera platform was adjusted to align the image of each CR to the center of the corresponding camera's image (CR$_1$, CR$_2$; *red*). This aligned both camera axes with the center of corneal curvature (**C**), and ensured that the two cameras were equidistant from the eye, and therefore had the same image magnification. The distances $\Delta_1$ and $\Delta_2$ from the CR to the center of the pupil (**P**) in each camera's image were used to compute the angular position of the pupil relative to the cameras ($\theta_1$, $\theta_2$) (see Materials and methods, *Equation 9*). (**B**) Example camera frames showing the software-detected pupil and CR locations for each camera. CR$_1$' and CR$_2$' (*grey*) are reflections generated by the infrared LED above the opposite camera, and are ignored. Other abbreviations as in (**A**). (**C**) Validation of dual-angle video-oculography. Eye position measured by using video-oculography is plotted against true angle as the camera system was rotated about the eye of two anesthetized mice.

DOI: https://doi.org/10.7554/eLife.29222.005

additional improvement (2.3% additional variance explained compared to the two-channel model, *Figure 4B,C*). All subsequent measurements of eye position were derived from the single best channel of the magnetic sensor.

An alternative method of calibration can be used in applications in which eye velocity is the most relevant measure, rather than eye position. Eye position signals obtained from the magnetic and video systems were each differentiated, saccades and motion artifacts were removed, and the resulting velocity traces were each fit with a sine wave (see Materials and methods). The ratio of the amplitudes of the sine wave fits for the video and magnetic signals provided a calibration factor that could then be used to convert the differentiated magnetic sensor voltage into eye velocity (*Figure 4D*). This method has the advantage of not requiring precise temporal alignment between the video and magnetic recordings, since the amplitude of each signal is determined independently. Calibration factors obtained using the velocity method and the position method were similar in each mouse (p=0.55, paired *t*-test, n = 24).

To assess the robustness of magnetic eye tracking over time, eight mice were repeatedly calibrated each day for a week, using the velocity method. The reciprocal of the calibration factor represents the in vivo sensitivity of the magnetic sensor in mV/° (the slope in *Figure 4B*, *left*). On average, the sensitivity of the magnetic sensor decreased by only 4.2% after one week (*Figure 4E*, *left*). Even after many months, the magnetic implants were remarkably robust. In a population of ten mice recorded 5–7 months after implantation, only two had unusable eye movement signals. In the

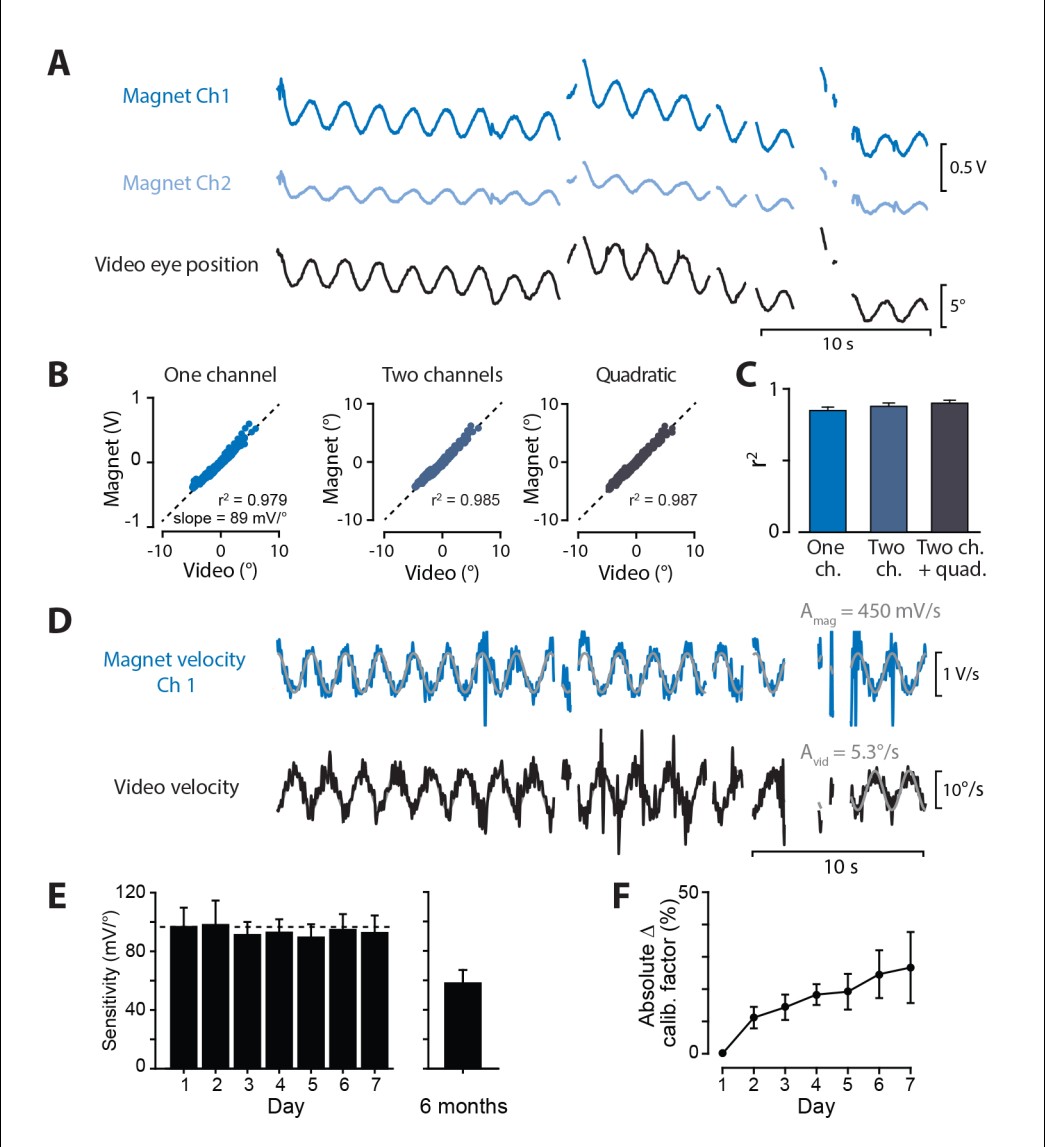

**Figure 4.** Calibration of magnetic eye tracking using video-oculography. (**A**) Simultaneously recorded magnetic sensor output (Ch1 *dark blue*, Ch2 *light blue*) and video-derived eye position (*black*) from one example mouse, measured during vestibular stimulation in the light. (**B**) Signals from the magnetic sensor, compared time point-by-time point against eye position measured by video-oculography, for the example mouse in (**A**). Results are shown for models of increasing complexity, from *left* to *right*: inputs to the regression model included the single best magnetic sensor channel alone ('*One channel*'), both magnetic sensor channels ('*Two channels*'), or two channels plus quadratic terms ('*Quadratic*') (see Materials and methods). (**C**) Mean correlation coefficients between video and calibrated magnet output for each model (n = 24 mice). (**D**) Calibration of magnetic sensor using eye velocity. Example differentiated video and magnetic sensor traces are shown for same mouse as in (**A**). The sine wave fit to the magnetic sensor channel with higher $r^2$ relative to the video is shown. The output of the magnetic sensor channel was scaled by a calibration factor equal to the ratio of video sine wave amplitude ($A_{vid}$) divided by magnetic sensor amplitude ($A_{mag}$). (**E**) Stability of magnetic eye tracking over time. Sensitivity was quantified as the reciprocal of the calibration factor obtained using the velocity method, and was measured each day for one week, and six months after magnet implantation. Mean sensitivity did not change over one week (all p>0.3, paired *t*-test, Days 2–7 vs. Day 1, n = 8). After six months (average time post-surgery 6.0 ± 0.4 months, range 5.2–7.7 months), sensitivity was lower but still robust (n = 8 mice; two additional mice had no detectable eye movement signals). (**F**) Changes in the calibration factor of individual mice over time. Although the mean sensitivity of the magnetic sensor was robust over months, the calibration factors of individual mice showed some fluctuations from day-to-

*Figure 4 continued on next page*

*Figure 4 continued*
day, as quantified by the absolute value of the percent change in calibration factor measured each day for one week, relative to Day 1.
DOI: https://doi.org/10.7554/eLife.29222.006

remainder, sensitivity was lower than on the first day it was measured, but still suitable for eye tracking measurements (*Figure 4E*, *right*).

Although the implants were robust over many months, the calibration factors could fluctuate from day to day in individual mice. To quantify this, the absolute value of the percent change in the calibration factor $k$, relative to the calibration factor value on Day 1, was calculated for each day in each mouse: $\Delta k_n = \frac{|k_n - k_1|}{k_1} \cdot 100\%$, where $k_n$ is the calibration factor on Day $n$. On average, there was gradual drift in the calibration factors from their original values over time ($\Delta k_n$ increased with the number of days since original calibration; *Figure 4F*), indicating that regular recalibration is important to maintain accuracy.

## Linearity and spatial resolution of the magnetic eye tracking system

We benchmarked magnetic eye tracking against the well-established eye coil technique, by quantifying the linearity and spatial resolution of each technique. Because the dual-angle video-oculography system reports eye position with a high degree of linearity (*Figure 3C*), the linearity of the magnetic and eye coil systems was assessed by regressing each signal against video-derived eye position, recorded simultaneously during vestibularly-driven eye movements in the light. Both eye coil and magnetic systems were highly correlated with the video system, and the correlation coefficient for the magnetic system was comparable to that for the eye coil system (*Figure 5A*), indicating a similar level of linearity.

The spatial resolution of the magnetic and eye coil systems was also compared. In head-fixed mice, spontaneous eye movements are interspersed with longer periods when the eyes are relatively stationary. The variability of the signals recorded during these stationary periods provides a measure of the eye tracking system's spatial resolution in vivo, with a lower bound determined by the actual jitter in eye position (*Stahl et al., 2000*; see Materials and methods). Quantified in this way, the magnetic and eye coil systems each had spatial resolutions of <0.1° (*Figure 5B*), similar to previous

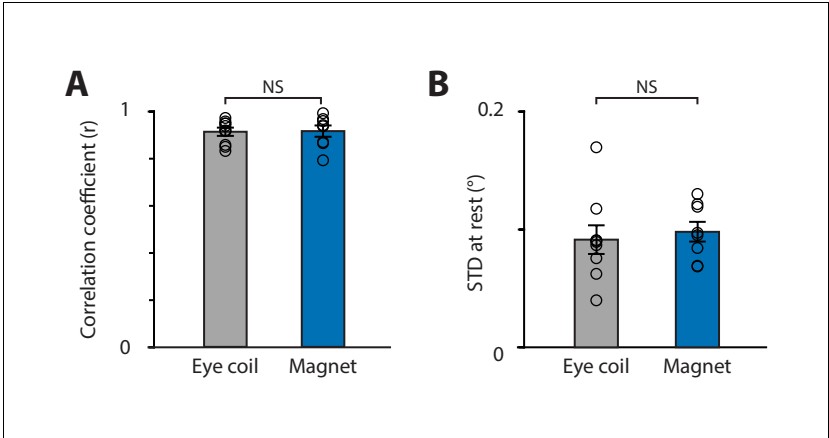

**Figure 5.** Comparison of magnetic eye tracking with the eye coil technique. (**A**) Correlation coefficient between the video and the eye coil data (*gray;* symbols represent individual mice, n = 9) and between the video and magnetic eye tracking data (*blue,* n = 8 mice) (p=0.937; two-sample *t*-test for eye coil vs. magnet). (**B**) Spatial resolution was similar for the eye coil and magnetic eye tracking systems (p=0.631, two-sample *t*-test), as measured by the standard deviation of eye position signals recorded during epochs when the eyes were relatively stationary in head-fixed mice in the light.
DOI: https://doi.org/10.7554/eLife.29222.007

reports for the eye coil (*Stahl et al., 2000*). Therefore, the spatial resolution, as well as the linearity, of magnetic eye tracking was comparable to that of the eye coil.

There has been controversy about whether eye coil implants alter eye movements in mice (*Boyden et al., 2006*; *Stahl et al., 2000*; *van Alphen et al., 2001*). To evaluate whether the magnet implant has any effect on eye movements, video-oculography was used to compare the eye movements measured one day before versus six days after implantation of the magnet and sensor. Eye movement responses to vestibular stimuli at a range of stimulus frequencies were measured in the dark. Because the eye movements driven by the vestibulo-ocular reflex (VOR) in the dark are open-loop, they should provide a particularly sensitive measure of any disruption of normal oculomotor dynamics. Neither the gain nor the phase of the eye movement responses were altered after magnet implantation (*Figure 6*). Likewise, the eye movements elicited by combined vestibular and visual inputs were unaltered by magnet implantation (*Figure 6—figure supplement 1*). The gain and phase of the VOR measured with magnetic eye tracking (*Figure 6*, *dashed, blue lines*) were in the range reported previously in mice by our laboratory (*Kimpo and Raymond, 2007*) and other laboratories (*Iwashita et al., 2001*; *van Alphen et al., 2001*), as were the gain and phase of the visually driven eye movements elicited by an optokinetic stimulus at a range of frequencies (*Figure 6—figure supplement 2*; *Iwashita et al., 2001*; *Kimpo and Raymond, 2007*; *Tabata et al., 2010*; *van Alphen et al., 2001*, *2009*, *2010*). Thus, magnet implantation had no detectable effect on oculomotor behavior.

## Bilateral eye movement recordings

The coordination of movements of the two eyes can influence visual experience. We assessed the suitability of magnetic eye tracking for simultaneously measuring the movements of both eyes of a mouse. Specifically, we tested whether there was interference created by the magnet in one eye influencing the sensor over the opposite eye. In three mice, a magnet was implanted in only one

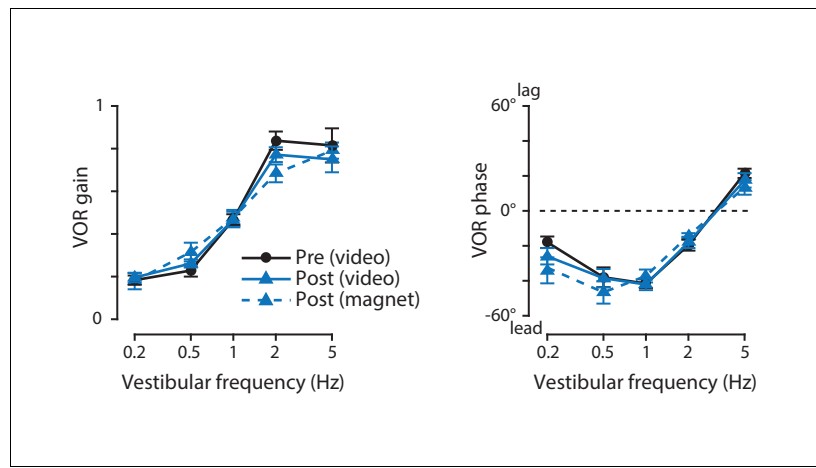

**Figure 6.** The vestibulo-ocular reflex (VOR) before and after magnet implantation. The gain (*left*) and phase (*right*) of the eye movement responses to vestibular stimuli, before (*black*) and six days after (*blue*) implantation of the magnet and sensor, measured using video-oculography with infrared illumination in otherwise complete darkness (*solid lines*). There was no effect of magnet implantation on either the gain or the phase of the VOR (p=0.725 gain, p=0.350 phase, for main effect of pre- vs post-implantation; p=0.497 gain, p=0.469 phase, for interaction with vestibular stimulus frequency; two-way repeated measures ANOVA, n = 8 mice). The gain and phase of the VOR measured simultaneously with magnetic eye tracking (post-implantation) are shown as well (*blue, dashed lines*).
DOI: https://doi.org/10.7554/eLife.29222.008

The following figure supplements are available for figure 6:

**Figure supplement 1.** Eye movements driven by combined vestibular and visual input, before and after magnet implantation.
DOI: https://doi.org/10.7554/eLife.29222.009
**Figure supplement 2.** The optokinetic response (OKR) measured with magnetic eye tracking.
DOI: https://doi.org/10.7554/eLife.29222.010

eye, while sensors were implanted above both eyes. The signals from both sensors were recorded during eye movements elicited by a sinusoidal vestibular stimulus (1 Hz, ±10°/s). The amplitude of the signal in the sensor contralateral to the eye with the magnet was only 2.7 ± 0.9% of that in the ipsilateral sensor, indicating limited crosstalk. Therefore, magnetic eye tracking allows simultaneous recording from both eyes, making it possible to analyze binocular coordination in mice.

## Eye movement recordings in freely moving mice

The small size and weight of the magnetic eye tracking system provided an opportunity to record eye movements in freely moving mice. Horizontal eye movements were recorded bilaterally for 5–10 min as mice explored a 12" arena surrounded by a stationary, vertically-striped visual stimulus. The eye movements recorded in this freely moving condition were compared with the eye movements recorded in the same mice while they were restrained in the center of the arena using their surgically implanted headpost. Overall, mice moved their eyes more when unrestrained, as evident in raw eye position traces (*Figure 7A*) and in scatterplots of eye position and eye velocity (*Figure 7B,D*) from individual mice. On average, freely moving mice displayed a two-fold increase in the standard deviation of eye position (*Figure 7C*), as well as a four-fold increase in the mean speed of the eye (*Figure 7E*) compared to head-fixed mice.

The increase in eye movements observed in freely moving mice was associated with active body movement, rather than simply the absence of restraint. Eye movements recorded in unrestrained mice were subdivided according to whether the body was actively moving or relatively still, as quantified by a motion index extracted from overhead video (see Materials and methods). During active, self-generated body movements, the mean eye speed was higher compared to when the body was relatively still (*Figure 7E*).

The correlation of eye movements with body movements did not simply reflect a general increase in motor activity or arousal, but were, at least in part, directly driven by motion of the head. Head movements create both vestibular input and visual input (from relative motion of the earth-stationary visual surround), which are both known to drive eye movements in mice (*Figure 2B,C*; *Iwashita et al., 2001*; *Katoh et al., 2008*; *Shin et al., 2014*; *van Alphen et al., 2001*). More specifically, head rotations about the yaw axis could drive the horizontal eye movements detected by our magnetic eye tracking system. Head movements were recorded in unrestrained mice using an inertial measurement unit. There was a strong, linear relationship between self-generated head movements about the yaw axis and horizontal eye movements in the opposite direction (*Figure 7F*), suggesting that active head movements are a major driver of eye movements in freely moving mice.

Not only were the motion statistics of individual eyes altered in freely moving mice, but the binocular coordination was altered as well. Binocular coordination is likely to play a key role in the active control of vision in most species. Such coordination is currently best understood in head-fixed primates, with much less known in lateral-eyed, afoveate, and freely moving animals, where there can be important differences (*Carriot et al., 2014*; *Land, 2015*), as illustrated by a recent study in rats (*Wallace et al., 2013*). We found that the eye divergence (the horizontal angular distance between the two eyes) increased in unrestrained mice, compared to during head fixation (*Figure 7G*, *left*), and the variability of eye divergence also increased (*Figure 7G*, *right*). Thus, bilateral eye movements in mice are highly dependent on the motion of the animal, with the eyes both more active and more divergent in freely moving compared to head-fixed mice.

## Discussion

Magnetic eye tracking opens up exciting new scientific opportunities by providing a high-resolution (<0.1°) method for measuring eye movements that overcomes the challenging physical constraints of small and freely moving animals. The methods described here extend previous reports of magnetic eye tracking (*Rodríguez et al., 2001*; *Salas et al., 1999*; *Schwarz et al., 2013*) by miniaturizing the implant for use in mice, increasing the resolution, assessing tolerance for various magnet-sensor alignments, and improving the calibration procedure. Magnetic eye tracking has important advantages over the commonly used eye coil and video-oculography techniques, and hence could replace these older techniques in many current applications. We developed the magnetic eye tracking system in mice due to the widespread use of this species, but there are no obvious barriers to extending this technique into larger species, such as rats or monkeys. A larger eye and orbit may require

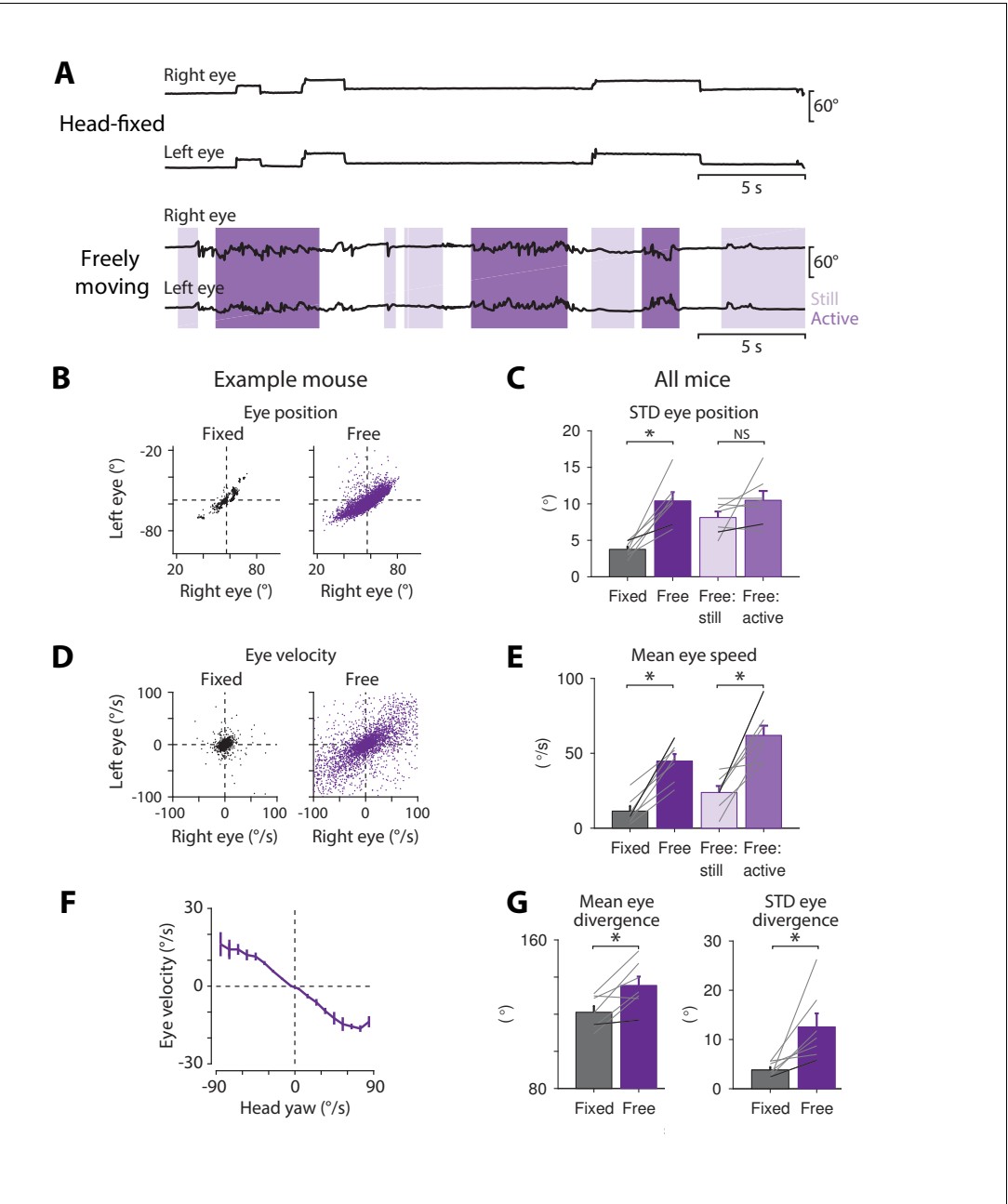

**Figure 7.** Eye movements measured in freely moving mice. (**A**) Example of bilateral eye movements in one mouse when it was head-fixed (*top*) compared to freely moving (*bottom*). *Light purple*, body still; d*ark purple*, body actively moving, as measured using overhead video images (see Materials and methods). (**B**) Scatterplot of left and right eye positions in the example mouse from (**A**), during head fixation (*black, left*) and during free motion (*purple, right*). (**C**) Variability of eye position in head-fixed and freely moving mice. The standard deviation of eye position increased in head-free (*dark purple*) compared with head-fixed (*black*) mice (p=1.9 × 10$^{-4}$, paired *t*-test, n = 7). Eye movements recorded in head-free mice were further subdivided into periods when the body was still (*light purple*) versus actively moving (*medium purple*) (p=0.146, paired *t*-test, n = 7). *Grey lines*, individual mice; *black lines*, example mouse in (**A**). (**D**) Scatterplots of velocity of the left and right eye during head fixation and free motion, for the example mouse in (**A**). (**E**) Mean eye speed was higher in freely moving than head-fixed mice (p=9.7 × 10$^{-5}$, paired *t*-test, n = 7). In unrestrained mice, eye speed was highest during periods of active body movement compared to periods when the body was relatively still (p=3.6 × 10$^{-4}$, paired *t*-test, n = 7). (**F**) Correlation between horizontal eye velocity and head velocity about the yaw axis in freely moving mice (n = 4). (**G**) Binocular coordination in head-fixed and freely moving mice. Left, mean eye divergence (horizontal angle between

*Figure 7 continued on next page*

*Figure 7 continued*

the two eyes) increased during free motion compared to head fixation (p=0.029, paired t-test, n = 7). Right, the standard deviation of eye divergence also increased during free motion (p=0.009, paired t-test, n = 7).

DOI: https://doi.org/10.7554/eLife.29222.011

that the sensor be implanted farther away from the magnet, however, this should be offset by the ability to implant a larger magnet.

## Comparison to the eye coil

Magnetic eye tracking achieved performance similar to that of the eye coil technique, which is widely considered the gold standard for measuring eye movements (*Robinson, 1963*). The spatial resolution of magnetic eye tracking (0.098°) was comparable to the resolution of the eye coil technique in mice, as measured by our lab (0.091°; *Figure 5B*) and other labs (0.09°, *Stahl et al., 2000*), and was also comparable to the resolution of the eye coil technique in larger species such as monkeys (<0.1°, *Kimmel et al., 2012*) and humans (0.096°, *Houben et al., 2006*). Further, the linearity of magnetic eye tracking was indistinguishable from that of the eye coil technique (*Figure 5A*).

The performance of the magnetic eye tracking system benefited from the use of angular magnetic field sensors containing anisotropic magnetoresistive (AMR) technology (*Schwarz et al., 2013*). In contrast to Hall effect sensors, which measure magnetic field strength (*Rodríguez et al., 2001*; *Salas et al., 1999*), AMR sensors are designed to measure the angle of a magnetic field. AMR sensors can achieve high resolution, while satisfying the constraints of low power, low cost, and small sensor size (*Lenz and Edelstein, 2006*). Despite the small size of the magnet required for mice, high resolution was achieved by placement of the magnet and sensor within the range established by systematic testing outside the animal.

With comparable spatial resolution, magnetic eye tracking offers several advantages over the eye coil technique. Most notably, the minimal equipment required for magnetic eye tracking permits its use in freely moving animals, as discussed further below. Magnetic eye tracking is also far less expensive than commercially available eye coil systems, making this approach available to more investigators. A third key advantage of magnetic eye tracking is the lack of fragile wires leading to the eye, which minimizes the difficulty of surgical implantation, greatly reducing the time required for the experimenter to learn and perform the procedure. The lack of fragile wires also extends the long-term reliability of the magnet implants. Only two out of ten mice followed for half a year after surgery had failed magnet implants, whereas the wires used for eye coil recordings often fail after several weeks following implantation (*Stahl et al., 2000*; Raymond lab, unpublished observations). Improved reliability allows for larger sample sizes with less attrition, which is particularly important when using valuable transgenic mice.

A concern has been raised that the weight of an eye coil or the presence of wire leads might physically impair eye movements in mice (*Stahl et al., 2000*). We found no evidence for an effect of magnet implantation on oculomotor performance (*Figure 6*; *Figure 6—figure supplement 1*). However, we do find that the experience and skill of the surgeon improves the outcome of both magnet and eye coil implantation surgeries in mice.

We replicated the general finding of an increase in VOR gain and decrease in OKR gain as well as an increase in VOR and OKR phase lag with stimulus frequency, which is observed across species. The gain and phase of the VOR and OKR measured with magnetic eye tracking fall within the range reported previously in mice using video and eye coil techniques, although there is considerable variability in the previously reported values, arising, at least in part, from sensitivity to the parameters of the vestibular and visual stimuli used to elicit eye movements, as well as behavioral factors related to handling and stress (*Iwashita et al., 2001*; *Kimpo and Raymond, 2007*; *Tabata et al., 2010*; *van Alphen et al., 2001*, *2009*, *2010*).

Magnetic eye tracking has many advantages, but also some disadvantages relative to the eye coil method. One current limitation is that the magnet-sensor configuration we used measures angular eye position about only one axis. A single axis is sufficient for many applications (*Iwashita et al., 2001*; *Kimpo et al., 2014*; *Koyama et al., 2004*; *Medrea and Cullen, 2013*; *Prusky et al., 2004*; *Wurtz, 1968*), moreover further development to incorporate either a second sensor or a 3D

magnetic field sensor should enable simultaneous measurement of eye movements along multiple axes simultaneously. A second disadvantage of magnetic eye tracking is that it requires a separate calibration device, whereas the eye coils can be calibrated by rotating the external magnetic field coils relative to the mouse. Early efforts at magnetic eye tracking employed an invasive calibration method involving physical rotation of the eye using a needle under a microscope (*Rodríguez et al., 2001*; *Salas et al., 1999*). We have minimized the difficulty of the calibration step by developing a rapid (<10 min), accurate, noninvasive calibration procedure using an improved video-oculography method, which can be performed in awake mice.

## Dual-angle video-oculography

Standard video-oculography approaches are challenging in mice and other afoveates because of the difficulty in estimating $R_p$ (the distance from the plane of the pupil to the center of corneal curvature), which is typically used to determine angular eye position from video images. $R_p$ has been approximated as the average radius of the eye in a species (*Katoh et al., 1998*; *Mangini et al., 1985*; *Schwarz et al., 2013*). However, this is an imperfect measure of $R_p$ because the curvature of the cornea is different from the rest of the eye, and because the eye does not rotate exactly about its center (*Stahl et al., 2000*). In addition, a single $R_p$ value doesn't account for variation across age, gender, genotype, or individuals (*Stahl et al., 2000*; *Wisard et al., 2010*). Methods have been devised for estimating $R_p$ in individual animals (*Stahl et al., 2000*; *Zoccolan et al., 2010*), but these methods extend the time the animal must be head-restrained for an experimental session. Most notably, the above methods assume that $R_p$ is constant over time, whereas $R_p$ is known to change rapidly with shifts in arousal or light intensity (*Kaufman, 2002*; *Kimmel et al., 2012*; *McCamy et al., 2015*; *Reimer et al., 2014*), which introduces inaccuracies.

Dual-angle video-oculography eliminates the need to estimate $R_p$, by calculating eye position from frame-by-frame comparison of the images from the two cameras. The result is highly linear and sufficiently accurate for the purpose of calibrating the magnetic eye tracking system. If enhanced with high-speed cameras and faster image processing, the dual-angle method could also stand alone as a powerful video-oculography technique.

Despite the improvements in video-oculography achieved by the dual-angle technique, magnetic eye tracking provides key advantages that make it preferable to video-oculography for many applications. First, the spatial resolution of magnetic eye tracking (0.1°) is superior to what has been reported for video-oculography in rodents (0.23°, *Stahl et al., 2000*; ~1°, *Wallace et al., 2013*), in monkeys (video resolution 2–3 × worse than eye coil, *Kimmel et al. (2012)*; *McCamy et al., 2015*) and in humans (0.56°, *Houben et al., 2006*). Second, magnetic eye tracking achieves high temporal resolution with less expense. Although some video-oculography systems can sample at 1,000 Hz, they are costly and typically sacrifice image resolution. Since the dual-angle system is used only for calibration, high-speed cameras are not required. Third, removal of the bulky video-oculography equipment after calibration allows free, unobstructed interaction of the animal with the environment during eye tracking. Technological advances have reduced the size of cameras to fit on the head of a rat (*Wallace et al., 2013*), but the hardware required for magnetic eye tracking is more than an order of magnitude lighter (0.05 cm$^3$ and 0.180 g total) and is therefore trivial for a mouse to carry, allowing much greater mobility than possible with video-oculography (*Kretschmer et al., 2017*). Also, either the video cameras themselves, or the edges of infrared-reflective mirrors, are visible to the animal during video-oculography, limiting experimental control of the visual environment. Thus, magnetic eye tracking is preferable for experiments in which spatial resolution, low cost, unobstructed vision, or compatibility with free motion is a priority.

## Applications

Magnetic eye tracking makes it possible to obtain reliable, high-resolution measurements of eye movements in small, freely moving animals, thereby creating new opportunities for studies of sensation, motor control, and cognition under more natural conditions. Neuroscientists are increasingly interested in studying the neural basis of behavior under conditions where the animal is free to move, particularly in rodents (e.g., *Licata et al., 2017*; *Davidson et al., 2009*; *Kepecs et al., 2008*; *Erlich et al., 2011*; *Jaramillo and Zador, 2014*). Free motion allows animals to perform orienting movements, which are a key part of any animal's behavioral repertoire. During visually guided

behaviors, coordinated movements of the trunk, head, and eyes can direct the animal's gaze towards relevant visual stimuli (*Belton and McCrea, 1999*; *Erlich et al., 2011*; *Fang et al., 2015*; *Freedman and Sparks, 1997*; *Fuller, 1992*; *Mather and Baker, 1980*). Body location and head direction are routinely measured during visually guided behaviors such as navigation, and have been shown to be encoded by the activity of neurons in relevant circuits (*Fyhn et al., 2008*; *Hardcastle et al., 2017*; *Yoder and Taube, 2009*). However, the technical challenges associated with measuring eye movements have limited the analysis of gaze control in freely moving animals. Our finding of robust eye movements in freely moving mice (*Figure 7*) suggests that gaze direction deviates substantially from head direction, as previously reported in rats (*Wallace et al., 2013*). Therefore, gaze direction may be an additional variable represented and controlled by the relevant neural circuits.

The magnetic eye tracking system facilitates investigation of many open questions regarding the neural control of gaze, and its role in guiding the acquisition of visual information and control of visually-guided behaviors. Eye movements provide a window into many sensory, motor, and cognitive processes in healthy and diseased brains. Technical challenges have limited the application of this experimental measure in the increasingly common neuroscientific studies of freely moving rodents. Magnetic eye tracking provides a high-resolution, low cost, and easily implemented method, which can enable more investigators to capitalize on the power of eye movement analysis.

## Materials and methods

### Animals

Magnetic eye tracking and video-oculography were performed in 35 C57BL/6 mice and two hybrid C57BL/6 × 129S1/SvlmJ F2 mice. Eye coil and video-oculography recordings were made in an additional nine C57BL/6 mice. All procedures were approved by Stanford University's Administrative Panel on Laboratory Animal Care.

### Surgery

A plastic head post was secured to the skull with three screws and dental cement, for use in immobilizing the head. In most animals, a magnet was implanted in one or both eyes. Different magnets were assessed. Most mice were implanted with two neodymium magnets of size 0.75 × 1 mm (diameter × height, grade N50, axially magnetized, SuperMagnetMan.com), which were stacked end-to-end for a total size of 0.75 × 2 mm (weight: 6.8 mg). These magnets were coated in the biocompatible polymer parylene. The magnetic force firmly held the two magnets together, and the resulting shape is referred to as the 'cylinder magnet'. Cylinder magnets were implanted in a single eye in 24 mice, and bilaterally in an additional seven mice. Mice were excluded from the analysis if reliable video-oculography was not possible following implantation due to scarring on the surface of the eye, or if an eye movement gain of less than 0.1 was recorded by the magnetic sensor during calibration, suggesting that the magnet-sensor alignment was inadequate. A 1.5 × 0.5 mm disc magnet was implanted in four additional mice, and performed similarly (data not shown). In pilot experiments, a smaller, 1.0 × 0.5 mm disc magnet was implanted in two mice, however, because the signals recorded using these magnets were very small (data not shown), we discontinued their use. Magnets were implanted following a procedure similar to that described previously for eye coil implantation (*Boyden and Raymond, 2003*). The conjunctiva was blunt dissected to form a pocket on the temporal side of the eye, slightly dorsal to the midline of the eye and anterior to the lateral rectus muscle. The magnet was inserted into this pocket using non-magnetic forceps (Dumoxel #5, FST) and the pocket was sealed shut with VetBond. To minimize any chance of obstructing or otherwise affecting vision, care was taken to implant the magnet well away from the cornea and to restrict the VetBond to the area surrounding the magnet. The cylinder magnet was implanted with the N-S axis aligned roughly perpendicular to the axis about which the eye rotates during horizontal (nasal-temporal) eye movements (*Figure 2A*). The disc magnet was implanted flat on the surface of the eye, with the N-S axis pointing towards the eye's center.

An angular magnetic field sensor (HMC1512, Honeywell Inc., size: 4.8 × 5.8 mm; weight: 76 mg) was secured to the skull above the eye using dental cement anchored to the same skull screws as were used to secure the head post. The HMC1512 sensor converts the direction of an external

magnetic field into an analog voltage output. The magnetic field is sensed by anisotropic magneto-resistive (AMR) elements, which change their electrical resistance according to the angle $\theta_m$ of an applied magnetic field. Each set of AMR elements is configured as a Wheatstone bridge to produce a differential voltage proportional to $\sin(2\theta_m)$. The HMC1512 contains two sets of AMR elements, rotated by 45° relative to each other.

The magnetic sensor was affixed to the skull above the implanted magnet, with its surface in a plane roughly parallel to the nasal-temporal axis of the eye and to the N-S axis of the implanted magnet (*Figure 2A*). This positioning allowed detection of horizontal eye movements. The sensor was positioned as close as possible to the eye (2.7 ± 0.2 mm, n = 8 mice, distance from the bottom surface of the sensor to the nasal-temporal axis defined by the two corners of the eye).

In seven mice, magnets and magnetic sensors were implanted in both eyes. To assess cross-talk between a magnet in one eye and the sensor above the opposite eye, three of these mice were first implanted with a magnet in the left eye alone, with sensors above both eyes. A magnet was later implanted in the right eye as well.

In nine mice, eye coil implantation was performed as described previously (*Boyden and Raymond, 2003*). Recordings from all mice were made after at least five days of recovery from surgery.

## Data acquisition

Differential outputs from the two magnetic sensor channels were amplified 60x by a custom pream-plifier (David Profitt Engineering Services, Los Altos, CA), then digitized and stored using a Power1401 A-D converter with Spike2 software (Cambridge Electronic Design, Cambridge, England). Connections from the sensor to the amplifier were formed via an 8-pin right angle connector (Digi-Key: 609-3694-1-ND, pins; 609-3705-1-ND, socket), which could be rapidly soldered to the sensor without the need for extra wiring (total weight of sensor, solder, and connector: 180.4 mg). Magnet and eye coil data were digitally filtered with an identical 100 Hz low-pass Butterworth filter.

## Magnetic sensor testing outside the animal

Performance of different magnet-sensor configurations was first assessed outside the mouse. Magnet angle was precisely controlled using a servo-controlled turntable (angular resolution 0.053°, standard deviation at rest), and the position of the magnet relative to the sensor was controlled with a micromanipulator (*Figure 1*). Repeated measurements were performed by removing the magnet from the apparatus and then repositioning it. Signal strength was compared across magnet-sensor configurations by calculating the sensitivity of the magnetic sensor to the angular position of the magnet, in mV/°. For each channel of the magnetic sensor, sensitivity was averaged in a ± 22.5° range around each zero crossing of that channel's output.

## Magnetic eye tracking in head-fixed mice

To assess performance of the magnet-sensor system after implantation in mice, eye movements were elicited with vestibular and visual stimuli in head-fixed mice. The head of the mouse was restrained, via the surgically implanted head post, in a natural position with the nose tilted down approximately 30° relative to the stereotaxic plane, which put the nasal-temporal axis of the eye in a roughly earth-horizontal plane. To elicit horizontal eye movements, vestibular stimuli were delivered by a servo-controlled turntable (Ideal Aerosmith, Inc.) that rotated the mouse about an earth-vertical axis. During the freely moving experiments (*Figure 7*), the stationary drum surrounding the animal had black and white stripes, each subtending 7° of visual angle. For OKR measurements (*Figure 6—figure supplement 2*), the drum was patterned with a random checkerboard, each square colored black or white with 50% probability and subtending 3° of visual angle. The vestibular and visual stimuli had sinusoidal motion profiles, at frequencies of 0.2, 0.5, 1, 2, or 5 Hz, with ±10°/s peak velocity. The gain of the eye movements driven by the vestibular and visual stimuli was calculated by dividing the amplitude of a sinusoidal fit to the eye velocity response by the amplitude of the stimulus (10°/s).

## Magnetic sensor calibration

Magnetic eye tracking requires calibration to convert the raw voltage signals from the magnetic sensor into eye position in degrees. Calibration was performed for each mouse by comparing signals from the magnetic sensor with eye movements recorded using a dual-angle video-oculography

system developed for this purpose and described below. Once calibrated, the magnetic sensor could then be used alone, without the video system, allowing for testing in complete darkness, with higher temporal resolution, and while the mouse was freely moving.

During a calibration session, video frames and magnetic sensor voltages were collected simultaneously while eye movements were elicited in awake, head-fixed mice using a sinusoidal vestibular stimulus in the light (0.5 Hz or 1 Hz, ±10°/s peak velocity; *Figure 4*). Video frames were analyzed as described below to extract horizontal eye position. The vestibular stimulus and raw voltage traces from the magnetic sensor were recorded in Spike2 (CED, Cambridge, England). An infrared LED, driven by a TTL signal generated and recorded in Spike2, delivered 30 ms flashes of light at 1 Hz during the calibration session for use in aligning video and magnet traces.

Eye position data derived from the video recordings were upsampled to match the magnetic sensor data, and then differentiated by calculating the slope in a 50 ms sliding window to obtain eye velocity. Uncalibrated magnet data were differentiated in the same manner. An automatic velocity-threshold algorithm (video threshold: 50°/s, magnet threshold: 5 mV/ms) was used to detect saccades and motion artifacts in the video and magnet data. Time segments that were marked for removal in either the video data or the magnet data were excluded from both traces during further analysis.

The signals from the magnetic sensor were calibrated by comparing them with either eye position or eye velocity signals derived from the video recordings. To calibrate using eye position, the eye position recorded by the video was linearly regressed on the voltage signals from the magnetic sensor. Three possible choices for the set of explanatory variables in the regression were compared (*Figure 4B,C*).

The simplest option was to estimate eye position using each of the two magnetic sensor channels, one at a time:

$$\hat{e}_1(t) = k_{1,1}m_1(t) + k_{2,1} \ and \ \hat{e}_2(t) = k_{1,2}m_2(t) + k_{2,2} \tag{1}$$

where $\hat{e}_1(t)$ and $\hat{e}_2(t)$ are the calibrated eye positions estimated fromy each channel, $m_1(t)$ and $m_2(t)$ are the voltages recorded by the two magnetic sensor channels, and $k_{1,1}$, $k_{1,2}$, $k_{2,1}$, $k_{2,2}$ are the model parameters. $k_{1,1}$ and $k_{1,2}$ correspond to the calibration factor for each channel, and $k_{2,1}$ and $k_{2,2}$ to the offset. The channel that was more strongly correlated with the video-derived eye position was then selected for further analysis.

A second method for calibrating the eye position-related signals from the magnetic sensor was to fit the video-derived eye position using both magnetic sensor channels simultaneously:

$$\hat{e}(t) = k_1 m_1(t) + k_2 m_2(t) + k_3 \tag{2}$$

To avoid overfitting, given the likelihood of correlated signals between the two channels, an L2 regularization penalty was applied to penalize model parameters with large weight (ridge regression). Ten-fold cross-validation was used to select the regularization parameter that minimized squared error on the held-out data. Once the regularization parameter was thus obtained, the regression was then run on all calibration data to yield the final model parameters $k_1, k_2$, and $k_3$.

A third method for converting the signals from the magnetic sensor to an estimate of eye position included quadratic terms, to try to improve the fit should the eye position fall in a non-linear portion of the output from both channels of the sensor:

$$\hat{e}(t) = k_1 m_1(t) + k_2 m_2(t) + k_3 m_1(t)^2 + k_4 m_2(t)^2 + k_5 \tag{3}$$

This quadratic regression was cross-validated with ridge regression as described above.

The recordings from the magnetic sensor could also be calibrated using signals related to eye velocity rather than eye position. The video and magnetic sensor signals related to eye position were differentiated using a 50 ms sliding window and then fit by sine waves, yielding a video-derived signal amplitude $A_{vid}$ (in units of °/s), and magnetic sensor-derived amplitudes $A_{mag}$ (in units of mV/s) for each sensor channel. The sensor channel with the higher $r^2$ from the fit to a sine wave was selected, and its differentiated output $\dot{m}(t)$ was converted to eye velocity $\hat{\dot{e}}(t)$ using a calibration factor obtained from the ratio of $A_{vid}$ and $A_{mag}$:

$$\hat{e}(t) = \frac{A_{vid}}{A_{mag}} \dot{m}(t) \tag{4}$$

Overall, the calibration procedure added less than 10 min, and typically less than 5 min, to the experiment. The procedure includes placing the mouse in the head restraint (1–2 min), positioning the video cameras (1–3 min), data collection (1–3 min), and adjustment of video-oculography software parameters (0–2 min). Once calibration was complete, the video cameras were removed from the experimental setup.

### Analysis of spatial resolution

To compare the spatial resolution of the magnetic eye tracking and eye coil techniques, calibrated eye position signals were recorded while the mouse was head-fixed in the light. Two-second epochs when the eye movements were minimal were selected for analysis. Spatial resolution was calculated as the standard deviation of eye position during each stationary epoch, with three epochs selected and averaged for each mouse.

### Magnetic eye tracking in freely moving mice

Seven mice implanted bilaterally with magnets and sensors were allowed to freely explore a 12" circular arena for 10–15 min (*Figure 7*). The stationary arena was surrounded by a stationary, illuminated, vertically-striped drum. To quantify whether the mouse's body was actively moving or relatively still, an overhead video acquired infrared images of the mouse in the arena. A motion index was calculated by first subtracting an average background image from all frames, and then quantifying the mean absolute difference in pixel values between each frame and the next. The resulting motion trace was smoothed with a 2 s moving average to create a motion index. The upper third of motion index values were classified as 'Active', and the lower third were classified as 'Still'. Eye position variability and mean eye speed measured in both eyes of a given mouse were first averaged together before comparing the population means.

To measure head movements in freely moving mice, a lightweight (1.8 g) six degree-of-freedom inertial measurement unit (SEN-10121, Sparkfun Electronics) was attached to the mouse's headpost with a screw. Roll, pitch, and yaw angles were converted into analog signals using an Arduino Mega (Arduino AG) and recorded simultaneously with eye position in Spike2.

### Dual-angle video-oculography

We developed a dual-angle video-oculography system for use in calibrating the magnetic eye tracking system. This system was designed to improve upon previous video-oculography systems in terms of accuracy, cost, and speed of use. The infrared-blocking filter was removed from two web cameras (Logitech C310) and replaced with a visible light-blocking filter. Small infrared LEDs (875 nm, radiant power 20 mW, angle of half intensity 20°, TSHA4400, Digi-Key) were positioned above each camera to create two corneal reflections (CRs), one for each camera, appearing along the radius of the cornea parallel to the axis of the camera.

The dual-angle system depends on two conditions. First, the two cameras must be fixed at a known angle relative to each other, for use in the calculation below. Second, they must be equidistant from the eye, so that the images from the two cameras have the same spatial scale. To achieve these requirements, the cameras were fixed to a platform separated by an angle of 40°, with each lens equidistant (in our setup, A = 5 cm) from a small placeholder target. Correct positioning was confirmed by verifying that the image of the target was the same size in the two cameras, and appeared at the horizontal midpoint of each camera's image. For use in mice, the placeholder target was removed, and the entire platform was positioned so that the pupil appeared roughly centered between the two CRs, to ensure that the pupil remained fully in the image when the eye rotated. The position of the platform was further adjusted so that for each camera, the CR generated by that camera's infrared LED appeared at the center of the h camera's image (*Figure 3B*), and the camera platform was then fixed in place. This ensured that the center of the eye's corneal curvature was close to the point at which the two camera axes intersected, thus placing each camera equidistant from the eye as required by the configuration of the platform.

The pupil was visualized by illuminating the cornea with infrared light, which causes the pupil to appear dark (*DiScenna et al., 1995*; *Kaufman, 2002*; *Stahl et al., 2000*). Sufficient contrast of the pupil relative to the cornea was provided by the two infrared LEDs used to create the CRs. Most measurements with video-oculography were conducted with additional, visible light, which constricted the pupil sufficiently to provide a clear image of pupil boundaries. A few video-oculography experiments were conducted in low light conditions, with only the two infrared LEDs (*Figure 6*), which caused the pupil to dilate and become occluded by the eyelid. To constrict the pupil for these recordings, a single drop of 4% pilocarpine (Henry Schein, Melville, NY) was placed on the eye 3–5 min before the experiment.

Images were acquired in MATLAB (The MathWorks, Inc., Natick, MA) at 30 frames per second, and were analyzed offline. The locations of the pupil center (P) and CRs in the images were extracted using modified code from OpenEyes, an open source eye tracking library (www.thirtysixthspan.com/openEyes). OpenEyes uses the Starburst algorithm to locate the pupil edges (*Li and Parkhurst, 2006*). We modified the code by employing a Hough transform, a feature extraction technique, to provide more accurate CR localization. The CRs were then masked before processing the pupil image to avoid false detection of pupil edges. Code is available at https://github.com/hpay/eyetrack (copy archived at https://github.com/elifesciences-publications/eyetrack).

The locations of the pupil and reference CRs in the images were used to calculate eye position by leveraging the geometry of the dual-angle setup. In each camera's image, motion of the pupil location is due to both rotation and translation of the eye relative to the camera. The component of this motion attributable to rotation alone can be isolated by subtracting the CR location from the pupil location, to yield the distances $\Delta_1$ and $\Delta_2$ (*Figure 3A,B*). Because the CR always appears on the radius of the cornea parallel to the axis of the camera, $\Delta_1$ and $\Delta_2$ are invariant to translations of the eye relative to the camera (*Stahl et al., 2000*).

The distances between the pupil and the reference CRs are related to angular eye position by:

$$\Delta_1 = R_p sin(\theta_1) \tag{5a}$$

$$\Delta_2 = R_p sin(\theta_2) \tag{6b}$$

where $\theta_1$ and $\theta_2$ are the angles from the axis of the respective camera to the axis of the pupil (the line connecting the center of the pupil (P) to the center of corneal curvature (C); $\theta_1$ and $\theta_2$ both defined to be positive as drawn in *Figure 3A*), and $R_p$ is the distance from the plane of the pupil to the center of corneal curvature (point C in *Figure 3A*). Taking the ratio of the two distances eliminates $R_p$:

$$\frac{\Delta_1}{\Delta_2} = \frac{sin(\theta_1)}{sin(\theta_2)} \tag{7}$$

Since $\theta_1 + \theta_2 = 40°$,

$$\frac{\Delta_1}{\Delta_2} = \frac{sin(40° - \theta_2)}{sin(\theta_2)} \tag{8}$$

Using the trigonometric identity for angle subtraction and rearranging yields a formula for calculating angular eye position:

$$\theta_2 = atan\left(\frac{sin(40°)}{\Delta_1/\Delta_2 + cos(40°)}\right) \tag{9}$$

In lieu of two cameras, a similar procedure could be implemented with a single camera, by using either angled mirrors or prisms to capture images of the eye from two distinct angles.

The video system was validated by anesthetizing two mice with ketamine (60 mg/kg)+dexmedetomidine (1 mg/kg) and physically rotating the two-camera platform via a mechanical arm that rotated about a vertical axis aligned with the mouse's eye. At each location, five frames were captured and the resulting measurements were averaged (*Figure 3C*).

## Statistical analysis

Mean ± SEM is reported for all confidence intervals. Statistical tests were conducted using Matlab. All tests were two-tailed. Two sample *t*-tests were used to determine if there was a difference between two groups. Paired *t*-tests were used to determine if a significant change occurred at different times or behavioral conditions within the same animal. To assess changes in oculomotor responses after magnet implantation, a two-way repeated measures ANOVA was conducted, with stimulus frequency and time (before vs. after magnet implantation) as factors (*Figure 6*). The significance level for all statistical tests was set at $p < 0.05$. The Lilliefors test (Matlab) was used to assess normality of the data. Based on this test, all data included in the statistical tests were deemed to have been sampled from a normal distribution.

## Acknowledgements

We gratefully acknowledge Jason Schwarz and Eric Knudsen for their technical assistance, Bob Schneevis for construction of custom apparatus, and Aparna Suvrathan, Pragna Gaddam, Matthew Rienzo, and Donald Payne for helpful discussion and comments on the manuscript. Funding was provided by NIH R21 EY026152 (JLR) and a grant from the Simons Foundation/SFARI (543031, JLR).

## Additional information

### Competing interests

Jennifer L Raymond: Reviewing editor, *eLife*. The other author declares that no competing interests exist.

### Funding

| Funder | Grant reference number | Author |
| --- | --- | --- |
| National Institutes of Health | R21-EY026152 | Hannah L Payne<br>Jennifer L Raymond |
| Simons Foundation/SFARI | 543031 | Jennifer L Raymond |

The funders had no role in study design, data collection and interpretation, or the decision to submit the work for publication.

### Author contributions

Hannah L Payne, Data curation, Software, Formal analysis, Funding acquisition, Investigation, Visualization, Methodology, Writing—original draft, Writing—review and editing; Jennifer L Raymond, Conceptualization, Resources, Supervision, Funding acquisition, Project administration, Writing—review and editing

### Author ORCIDs

Hannah L Payne ![ORCID] http://orcid.org/0000-0003-4625-5706
Jennifer L Raymond ![ORCID] http://orcid.org/0000-0002-8145-747X

### Ethics

Animal experimentation: All procedures complied with the recommendations in the Guide for the Care and Use of Laboratory Animals of the National Institutes of Health. Animals were handled strictly according the institutional animal care and use committee (IACUC) at Stanford University, accredited by the Association for Assessment and Accreditation of Laboratory Animal Care International (AAALAC). Protocols were approved by the Administrative Panel on Laboratory Animal Care at Stanford University (protocol #9143). All surgery was performed under isoflurane anesthesia, and every effort was made to minimize animal suffering.

Decision letter and Author response
Decision letter https://doi.org/10.7554/eLife.29222.014
Author response https://doi.org/10.7554/eLife.29222.015

---

## Additional files

### Supplementary files
• Transparent reporting form
DOI: https://doi.org/10.7554/eLife.29222.012

---

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
