## [Decision Letter]

Thank you for submitting your article "Magnetic eye tracking in mice" for consideration by *eLife*. Your article has been reviewed by three peer reviewers, one of whom, Geoffrey Schoenbaum (Reviewer #1), is a member of our Board of Reviewing Editors, and the evaluation has been overseen by Richard Ivry as the Senior Editor. The following individuals involved in review of your submission have also agreed to reveal their identity: Matt Gardner (Reviewer #2) and José María Delgado-García (Reviewer #3).

The reviewers and Reviewing Editor have discussed the reviews with one another and drafted a decision letter to help you prepare a revised submission.

Summary:

In this manuscript the authors present a novel method using magnetic sensing that allows them to track eye movement in mice. This is timely given the increasing interest and in fact implementation of visual and touchscreen based tasks in rodents.

Essential revisions:

1) Please confirm that there are no substantial effects on vision (or discuss this potential issue) and the requirement for head restraint. While this is a general statement, it is particularly relevant with respect to the start of subsection “Surgery”.

2) Is there any issue moving into rodents with the approach outlined? Discussion of this is relevant when considering potential impact of this methodological advance.

Secondary suggestions for revision:

1) A technique's usefulness is dependent on several characteristics such as the scope of application, the necessity of a solution, its effectiveness and its ease of use. The authors' have comprehensively demonstrated its effectiveness, and the scope and need for a solution are more up to the reader to consider. Nonetheless, the authors' should provide a bit more detail on the ease of use of the technique for researchers who might be considering whether to try it. The authors' state that the daily calibration only takes a short bit of time, yet does require head-restraint. Does the calibration require the mice to be anesthetized-- we expect not, but the authors should make this explicit. It would be useful to add details on the daily calibration procedure in reference to steps and time taken for the full procedure.

2) The comparisons of eye movements on the head retrained and freely moving mice are intriguing. Is it possible that some of this additional movement could be from forces which are not generated by the eye muscles themselves such as eye movement due to acceleration of the head? Of course this would still be eye movement, but the more interesting question that the authors appear to be addressing is whether there is increased eye movement initiated by the animal, not whether the additional forces which necessarily occur during free movement cause the effect. This might be hard to address, but should be brought up due to the claims made in the paper.

3) The low frequency shifts in Figure 2 are puzzling. Are these due to a slow shift in gaze, a measurement artifact or something else?

4) Given the many contributions of David Robinson to the study and understanding of the oculomotor system, including the precise quantification of eye motor responses and neural organization of the system, it would be appropriate to cite this work (especially since many of the references build on Robinson's contributions).

5) Subsection “Magnetic tracking of eye movements in mice" paragraph two and three. For comparative purposes with other recording systems and with other species, the gain and phase (in degrees) of evoked VOR and OKN should be quantified and represented. Indicating that "the magnetic sensor detected robust eye movements" is not enough!

6) Subsection “Calibration of magnetic eye tracking using dual-angle video-oculography" paragraph four. The system seems to be suitable for horizontal eye movements, but what about the vertical component? The point is that eye movements in a free moving mouse will probably be the result of angular (horizontal + vertical) eye movements.

7) Figure 2. Please provide information regarding gain and phase (in degrees) of recorded eye reflexes.

8) Figure 6 comparison with previous description of gain and phase (expressed in degrees) in mice would be appropriate.

9) One reviewer stated, "I think that Vetbond is not indicated for a direct contact with the eye."

---

## [Author Response]

*Essential revisions:*

*1) Please confirm that there are no substantial effects on vision (or discuss this potential issue) and the requirement for head restraint. While this is a general statement, it is particularly relevant with respect to the start of subsection “Surgery”.*

With respect to potential effects of the implant on vision:

1) As we now indicate in the Materials and methods (subsection “Surgery”), we minimize the chance for any effect on vision by surgically implanting the magnet away from the pupil, in a pocket of the conjunctiva on the lateral portion of the eye. Notably, the magnet implantation surgery involves less manipulation of the eye than the eye coil implantation surgery that is performed in many studies of vision.

2) We now include additional data showing that visually-driven eye movements are normal in mice implanted with a magnet and sensor. First, we show that the gain and phase of the eye movements driven by combined visual and vestibular stimuli, as measured with video-oculography, are the same before vs. after magnet and sensor implantation (Figure 6—figure supplement 1). Second, we show that the eye movement responses to a visual (optokinetic) stimulus alone are intact after magnet and sensory implantation (Figure 6—figure supplement 2). Eye movements are a fairly sensitive behavioral measure of visual processing. Although we cannot rule out the possibility that more complex visual tasks could be affected by the magnet implantation, the observation that visually-driven eye movements are normal suggests that vision is largely intact.

With respect to the requirement for head restraint during calibration:

A major advantage of the magnetic eye tracking technique is that, once the system is calibrated, measurements of eye movements can be made without head restraint. However, as pointed out by the reviewers, head restraint is still required during the brief calibration procedure. As detailed below and in subsection “Magnetic sensor calibration” of the revised manuscript, the duration of restraint required for calibration is only 5-10 min, and the implantation of a head post for use in restraint adds minimally to the effort and invasiveness of the surgery.

*2) Is there any issue moving into rodents with the approach outlined? Discussion of this is relevant when considering potential impact of this methodological advance.*

There are no obvious barriers to extending this technique to other species, such as rats or monkeys. A larger eye and orbit may require that the sensor be implanted farther away from the magnet, however, this should be offset by the ability to implant a larger magnet. We now include this point in the Discussion section.

*Secondary suggestions for revision:*

*1) A technique's usefulness is dependent on several characteristics such as the scope of application, the necessity of a solution, its effectiveness and its ease of use. The authors' have comprehensively demonstrated its effectiveness, and the scope and need for a solution are more up to the reader to consider. Nonetheless, the authors' should provide a bit more detail on the ease of use of the technique for researchers who might be considering whether to try it. The authors' state that the daily calibration only takes a short bit of time, yet does require head-restraint. Does the calibration require the mice to be anesthetized-- we expect not, but the authors should make this explicit. It would be useful to add details on the daily calibration procedure in reference to steps and time taken for the full procedure.*

To help potential users of the magnetic eye tracking technique assess the effort involved, we now provide additional detail about the time required for each step (subsection “Magnetic sensor calibration”). The calibration procedure includes placing the mouse in the head restraint (1 minute), positioning the video cameras (1-3 minutes), calibration data collection (1-3 minutes), and adjustment of video-oculography parameters (0-2 minutes). Overall, the calibration procedure adds less than 10 minutes to an experiment.

We also clarify that calibration does not involve subjecting the mice to anesthesia, by adding the word “awake” to the calibration description.

Finally, we note in the description of the implant surgery (subsection “Surgery”) that the addition of a headpost for the purpose of head restraint is a minor additional step during surgery. Anchoring the magnetic sensor directly to the skull using dental cement achieved the most stable recordings, so some form of skull implant is necessary-- adding a head post in addition to the sensor doesn’t increase the invasiveness of the procedure.

*2) The comparisons of eye movements on the head retrained and freely moving mice are intriguing. Is it possible that some of this additional movement could be from forces which are not generated by the eye muscles themselves such as eye movement due to acceleration of the head? Of course this would still be eye movement, but the more interesting question that the authors appear to be addressing is whether there is increased eye movement initiated by the animal, not whether the additional forces which necessarily occur during free movement cause the effect. This might be hard to address, but should be brought up due to the claims made in the paper.*

The magnetic eye tracking technique opens up the opportunity to study this and many other questions about the control of eye movements, as indicated in the Discussion (subsection “Applications”). In the present manuscript, our primary goal was to introduce and validate this new measurement technique, and to illustrate its potential for addressing the many open questions about eye movements in freely moving animals. As such, our study was not designed to address the specific question of how the passive properties of the eye plant contribute to eye movements in freely moving mice. However, we have measured eye movements during head acceleration in anesthetized animals (unpublished observations), and see little eye movement response. Hence, we expect that the eye movements observed in freely moving mice are, to a great extent, actively driven by neural commands to the eye muscles.

*3) The low frequency shifts in Figure 2 are puzzling. Are these due to a slow shift in gaze, a measurement artifact or something else?*

These low frequency shifts in eye position are not a measurement artifact, nor are they unique to measurements made with the magnetic eye tracking technique. Slow shifts in eye position are well documented in the oculomotor literature and are observed across species (Stahl, 2004; Koekkoek et al., 1997; Stahl and Thumser, 2014; Tan and Collewijn, 1991). Importantly, this drift is observed in studies that use the noninvasive video-oculography technique to measure eye movements (Stahl, 2004; Koekkoek et al., 1997), so is not a measurement artifact of the magnetic eye tracking system (see also Figure 4).

The signals controlling eye position are actively generated by the oculomotor circuitry. The “oculomotor integrator”, which holds the eyes at eccentric positions, is not a perfect integrator, hence the eyes tend to drift back toward null position. In mice, the oculomotor integrator has a short time constant, hence, drift is more pronounced (van Alphen et al., 2001). In the dark, this drift is also more pronounced because it is not opposed by visual feedback.

To avoid distracting the reader with this point, which is an important issue in oculomotor control, but is independent of the techniques used to measure eye movements, which are the focus of this manuscript, we have replaced some of the example traces in Figure 2 with traces that exhibit less position drift (still all from the same mouse).

*4) Given the many contributions of David Robinson to the study and understanding of the oculomotor system, including the precise quantification of eye motor responses and neural organization of the system, it would be appropriate to cite this work (especially since many of the references build on Robinson's contributions).*

We agree that the work of David Robinson has laid the foundation for quantitative analyses of the oculomotor system, and we have added another reference to his work.

*5) Subsection “Magnetic tracking of eye movements in mice “paragraph two and three. For comparative purposes with other recording systems and with other species, the gain and phase (in degrees) of evoked VOR and OKN should be quantified and represented. Indicating that "the magnetic sensor detected robust eye movements" is not enough!*

We did not quantify the gain and phase of the VOR and optokinetic response (OKR) at this specific point in the text, because we had not yet described the calibration procedure, which is necessary to calculate the gain. The example traces in Figure 2 are included to provide motivation for the development of the video calibration system. After describing the calibration procedure, we now quantify the gain and phase of the VOR measured with magnetic eye tracking in a population of mice in Figure 6 (blue, dashed lines). Moreover, we have conducted additional experiments to quantify the gain and phase of the OKR in a population of mice (Figure 6—figure supplement 2). The gain and phase we measured with magnetic eye tracking at frequencies from 0.2 Hz to 5 Hz is in the range reported previously by our lab and other labs using the video or eye coil techniques (Iwashita et al., 2001; van Alphen et al., 2001; Kimpo and Raymond, 2007), although the VOR and OKR gain and phase have been shown to vary considerably with factors such as the spatial and temporal frequency, amplitude, and contrast of the vestibular and visual stimuli used to elicit the eye movement responses.

*6) Subsection “Calibration of magnetic eye tracking using dual-angle video-oculography" paragraph four. The system seems to be suitable for horizontal eye movements, but what about the vertical component? The point is that eye movements in a free moving mouse will probably be the result of angular (horizontal + vertical) eye movements.*

The control of eye movements is organized largely along three orthogonal axes (horizontal eye movements about an earth vertical axis and vertical eye movements about two earth horizontal axes, oriented at 45° and 135° relative to the midsagittal plane). Many oculomotor labs, including ours, have focused on the neural signals controlling eye movements along a single axis. It should be trivial to measure eye movements along a different axis (vertical vs. horizontal) by rotating the magnet and sensor positions. With additional technical development, incorporating either a second sensor or a 3D magnetic field sensor, it also should be possible to measure eye movements along multiple axes simultaneously, as mentioned in the Discussion.

*7) Figure 2. Please provide information regarding gain and phase (in degrees) of recorded eye reflexes.*

We report quantitative information regarding the gain and phase of the vestibulo-ocular reflex (VOR) (Figure 6), the visually enhanced VOR (Figure 6—figure supplement 1), and the optokinetic response (OKR) (Figure 6—figure supplement 2) for a range of stimulus frequencies.

*8) Figure 6 comparison with previous description of gain and phase (expressed in degrees) in mice would be appropriate.*

There is considerable variability in previous reports of the gain and phase of the VOR and OKR in mice, measured with eye coil (van Alphen et al., 2001; Kimpo and Raymond, 2007) or video (Iwashita et al., 2001; van Alphen et al., 2010; van Alphen et al., 2009; Tabata et al., 2010) techniques. These values are known to depend on parameters of the vestibular and visual stimuli, including temporal frequency, amplitude, and visual stimulus spatial frequency and contrast, as well as the age and gender of the animals. It is likely that additional behavioral variables related to handling, stress, and experience of the mouse also affect the eye movement responses. The values we measure with the magnetic eye tracking and video techniques fall within the range reported previously with video and eye coil methods. Moreover, there are certain consistencies across all studies, including an increase in VOR gain and decrease in OKR gain, plus an increase in VOR and OKR phase lag with stimulus frequency, which we also observe with magnetic eye tracking. We now include a brief discussion of this in the final paragraph of subsection “Linearity and spatial resolution of the magnetic eye tracking system”.

*9) One reviewer stated, "I think that Vetbond is not indicated for a direct contact with the eye."*

Cyanoacrylate has been safely used on the eye in humans (Lal et al., 2015), rabbits (Ollivier et al., 2001), dogs (Fridman et al., 2010), chickens (Carey et al., 1996) and mice (Boyden and Raymond, 2003). It was approved by our IACUC for use in magnetic eye tracking. Care is taken to restrict application of VetBond to only the pocket in the conjunctiva immediately surrounding the magnet, and we have observed no significant adverse effects. We have added a line about this to the Materials and methods. The cyanoacrylate should be absorbed within 14 days (Ollivier et al., 2001).

References

Carey, J.P., Fuchs, a F. and Rubel, E.W., 1996. Hair cell regeneration and recovery of the vestibuloocular reflex in the avian vestibular system. *Journal of neurophysiology*, 76(5), pp.3301–3312.

Fridman, G.Y. et al., 2010. Vestibulo-ocular reflex responses to a multichannel vestibular prosthesis incorporating a 3D coordinate transformation for correction of misalignment. *JARO – Journal of the Association for Research in Otolaryngology*, 11(3), pp.367–381.

Lal, I. et al., 2015. Efficacy of conjunctival resection with cyanoacrylate glue application in preventing recurrences of Mooren’s ulcer. *British Journal of Ophthalmology*, pp.971–975.

Ollivier, F., Delverdier, M. and Regnier, A., 2001. Tolerance of the rabbit cornea to an n-butyl-ester cyanoacrylate adhesive (Vetbond). *Veterinary Ophthalmology*, 4(4), pp.261–266.

Stahl, J.S., 2004. Using eye movements to assess brain function in mice. *Vision Research*, 44, pp.3401–3410.

Stahl, J.S. and Thumser, Z.C., 2014. Flocculus Purkinje cell signals in mouse Cacna1a calcium channel mutants of escalating severity: an investigation of the role of firing irregularity in ataxia. *Journal of neurophysiology*, 112(10), pp.2647–2663.

Tan, H.S. and Collewijn, H., 1991. Cholinergic modulation of optokinetic and vestibulo-ocular responses: a study with microinjections in the flocculus of the rabbit. *Exp Brain Res*, 85(3), pp.475–481.